# The Damaging Effects of Long UVA (UVA1) Rays: A Major Challenge to Preserve Skin Health and Integrity

**DOI:** 10.3390/ijms23158243

**Published:** 2022-07-26

**Authors:** Françoise Bernerd, Thierry Passeron, Isabelle Castiel, Claire Marionnet

**Affiliations:** 1L’Oréal Research and Innovation, 1 Avenue Eugène Schueller, 93600 Aulnay sous Bois, France; claire.marionnet@rd.loreal.com; 2Department of Dermatology, CHU Nice, University Côte d’Azur, 151, Route de Ginestière, 06200 Nice, France; passeron@unice.fr; 3Research Center C3M, INSERM Unit 1065, University Côte d’Azur, 06200 Nice, France; 4L’Oréal Research and Innovation, 3 Rue Dora Maar, 93400 Saint-Ouen, France; isabelle.castiel@rd.loreal.com

**Keywords:** UVA1, oxidative stress, photoaging, photocarcinogenesis, photoimmunosuppression, hyperpigmentation, photoprotection

## Abstract

Within solar ultraviolet (UV) light, the longest UVA1 wavelengths, with significant and relatively constant levels all year round and large penetration properties, produce effects in all cutaneous layers. Their effects, mediated by numerous endogenous chromophores, primarily involve the generation of reactive oxygen species (ROS). The resulting oxidative stress is the major mode of action of UVA1, responsible for lipid peroxidation, protein carbonylation, DNA lesions and subsequent intracellular signaling cascades. These molecular changes lead to mutations, apoptosis, dermis remodeling, inflammatory reactions and abnormal immune responses. The altered biological functions contribute to clinical consequences such as hyperpigmentation, inflammation, photoimmunosuppression, sun allergies, photoaging and photocancers. Such harmful impacts have also been reported after the use of UVA1 phototherapy or tanning beds. Furthermore, other external aggressors, such as pollutants and visible light (Vis), were shown to induce independent, cumulative and synergistic effects with UVA1 rays. In this review, we synthetize the biological and clinical effects of UVA1 and the complementary effects of UVA1 with pollutants or Vis. The identified deleterious biological impact of UVA1 contributing to clinical consequences, combined with the predominance of UVA1 rays in solar UV radiation, constitute a solid rational for the need for a broad photoprotection, including UVA1 up to 400 nm.

## 1. Introduction

The three major wavebands of solar radiation reaching the Earth are made of ultraviolet (UV), visible light (Vis) and infrared rays (IR). They also represent the solar rays known for their impact on skin (Figure 1). The complete range of UV rays is composed of UVC wavelengths (100–280 nm), which are stopped by the ozone layer, UVB wavelengths (280–320 nm) and UVA wavelengths (320–400 nm). UVA rays can be further divided into shortwave UVA (UVA2, 320–340 nm) and longwave UVA (UVA1, 340–400 nm). Since UV penetration properties increase with higher wavelengths, UVB photons, the shortest photons reaching the Earth’s surface, are easily blocked by the ozone layer and meteorological conditions but also by the solar inclination, latitude, season and hour of the day [1,2,3]. In line with this, UVA are less impacted by these factors—particularly UVA1 rays, the longest UV photons, which represent approximately 75% of solar UVA radiation and up to 80% of the total UV reaching the Earth [3,4] (Figure 1). UVA1 exposure can also occur artificially during skin UVA1-phototherapy treatments or during sunbed tanning sessions with devices emitting mainly UVA1 [5,6].

Compared with UVB, UVA rays are less energetic, but due to their higher penetration properties, they can pass through a cloudy sky or even glass. At the skin level, UVA rays can pass through the epidermis and reach the lower dermis. Approximately 100 times more UVA reach the dermis than UVB photons [7]. Although the total UVA wavelengths domain in general has been under scrutiny in the last two decades, the attention has focused on longwave UVA1 more recently, evidencing their contribution in solar UV harmful effects. Their biological impact on human skin had long been underestimated because of their lower energetic properties. The mechanism of action of UVA1 relies on chromophores, which absorb UVA1 photons and can induce oxidative stress in all skin layers from the epidermis to the dermis. The generation of reactive oxygen species (ROS) leads to lipid peroxidation, protein changes and the formation of DNA photoproducts.

In line with their biological effects affecting all skin compartments, the contribution of UVA1 in clinical consequences such as human photocarcinogenesis [7,8,9], skin photoaging [10,11], immune suppression [8,12] and hyperpigmentation [13,14] is now well established.

The biological and clinical consequences known to be associated with UVA1 phototherapy and tanning bed use further support the data collected in solar exposure conditions. Moreover, in real life all year long, the skin is also exposed to other external factors, such as pollutants and Vis, which have been shown to induce combined effects with UVA1 rays [15].

In this review, we synthetize the effects of UVA1 on the skin at the molecular, cellular and tissue levels, as well as the cumulative and synergistic effects of UVA1 with pollution or Vis. Based on this knowledge, we will discuss the rationale for the need for a broad UV photoprotection, including UVA1, up to 400 nm to preserve skin health and integrity.

## 2. Effects of UVA from Sunlight

### 2.1. Chromophores and Reactive Oxygen Species

#### 2.1.1. Endogenous Sensitizers

In contrast with UVB, UVA radiation at doses corresponding to realistic sun exposure induces major oxidative stress in cells. UVA effects are mediated by numerous chromophores present in human skin that strongly absorb the UVA. Endogenous photosensitizers include a wide variety of chemical structures and involve different skin localizations and photochemical mechanisms of action [16]. Besides ROS formation due to electron leakage from the mitochondrial respiratory chain and the remodeling of plasma membrane lipid rafts, classes of skin chromophores, such as porphyrins, bilirubin, melanin and its precursors, flavins, pterins and B6 vitamins, have established or likely roles as sensitizers of photooxidative stress (See Box 1: Main UVA chromophores).

Box 1Main UVA Chromophores.
Porphyrins are powerful UVA photosensitizers, particularly protoporphyrin IX [17].Bilirubin is a lipophilic pigment formed by the photooxidative breakdown of heme from the hemoglobin upon the UVA irradiation of erythrocytes in the dermal capillaries of human skin [18].Melanin and melanin precursors can act as both skin photoprotectors and photosensitizers [19]. Pheomelanin is more reactive and phototoxic than eumelanin, generally considered as photoprotective [20].Flavins are involved in blue light and UVA photoreception, DNA photorepair and cellular phototoxicity [21,22].Pterins are highly fluorescent skin UV chromophores that potentially participate in photosensitization reactions [23,24].UVA-irradiated B6 vitamins have a phototoxic [25] as well as a potential protective [26] role against singlet oxygen (^1^O_2_) damage.


Urocanic acid (UCA) is also a skin chromophore and acts as a mediator of skin photo-immunosuppression (as detailed in Section 2.4.2). It is not clear if UCA is an endogenous sensitizer of photooxidative stress, because it can act as a triplet state and singlet oxygen (^1^O_2_)-sensitizer/quencher and can also scavenge hydroxyl radical (OH^.^).

Other potential chromophores could include Vitamin K, although little information is available on its role as a potential UVA photosensitizer in human skin. Tryptophan and tryptophan photooxidation products have been described to act as both UV-photosensitizers as well as potent physical and chemical quenchers of O2. Finally, nicotinamide adenine dinucleotide phosphate (NAD(P)H), an abundant fluorophore in skin that strongly contributes to skin autofluorescence upon UVA photoexcitation, does not seem to qualify as a relevant endogenous photosensitizer in human skin [16].

#### 2.1.2. Oxidative Stress: ROS Formation and the Release of Free Iron

UVA exposure can lead to the generation of ROS such as singlet oxygen (^1^O_2_), hydroxyl radical (OH^.^), superoxide anion (O_2_^−^) and hydrogen peroxide (H_2_O_2_) [27] but also mediates the release of the cellular labile iron pool [28] (Figure 2).

Photon absorption by chromophores in the electronic ground state induces the formation of chromophores in photoexcited states. The photoexcited sensitizer can react with molecular oxygen in type II reactions, leading to the formation of singlet oxygen (^1^O_2_) and superoxide radical anion (O_2_^−^). Superoxide radical anion (O_2_^−^) can also be produced after the direct reaction of the photoexcited sensitizer with substrates molecules (such as DNA bases) in a type I reaction. Superoxide radical anions (O_2_^−^) can then lead to H_2_O_2_ formation by dismutation. UVA radiation can also influence the levels of pro-oxidant catalysts, particularly leading to the release of free iron and heme iron in cells [29]. Iron plays a catalytic role in Fenton chemistry by undergoing a redox cycling between ferrous (Fe^2+^) and ferric (Fe^3+^) states in the presence of endogenous reductants and oxygen, which leads to hydroxyl radical generation in cells [30]. The activation of membrane-associated NADPH oxidase 1 (NOX-1 oxidase) in keratinocytes generates a swift increase in ROS, particularly superoxide anion (O_2_^−^).

In addition to ROS, skin photooxidative stress is also mediated by organic free radicals and other toxic photoproducts.

Oxidative stress is defined as an imbalance between free radicals and cellular antioxidant capacities.

#### 2.1.3. Cellular Defense

Levels of both pro-oxidant and antioxidant proteins are increased following moderate doses of UVA. On one side, damages to the heme-containing antioxidant enzyme catalase contribute to an altered redox state. On the other side, in order to restore the redox homeostasis, some cellular antioxidant mechanisms are elicited by UVA, such as the activation of the manganese-dependent superoxide dismutase (MnSOD) and glutathione peroxidase (GPx), which decrease the levels of superoxide and peroxide, respectively [30].

In response to oxidative stress, the transcription factor Nrf2 translocates to the nucleus and transactivates the expression of several dozen cytoprotective genes that increase cell survival, including detoxication and antioxidant enzymes [31,32]. Among them, the activation of heme oxygenase 1 (HO-1) also contributes to reducing the free heme level.

The involvement of the Nrf2 defense response to UVA1-induced oxidative stress was also evidenced using a reconstructed human skin model composed of both a living dermal equivalent and a fully differentiated epidermis presenting a three-dimensional architecture [33]. This model allows researchers to take into account UVA1 penetration properties and to collect transcriptomic data in epidermal keratinocytes and dermal fibroblasts. Two to six hours following the generation of ROS by UVA1 in this model, the expression of Nrf2 target genes such as HMOX1, TXNRD1, NQO1, FTL, GCLM, AKR1C2 and AKR1C3 was upregulated [33]. Moreover, several members of the chaperone molecules family of heat shock proteins (HSP) (HSP72, HSPB8, HSP70 and HSP40) were induced and could play a role as a natural defense mechanism against UV [33,34]. Nrf2 target genes were also upregulated in vivo, as shown in 62 volunteers from three different descents six hours after UVA1 exposure [13].

In conclusion, some cellular mechanisms can partially cope with oxidative stress, but these systems can be overwhelmed [30].

#### 2.1.4. Bystander Effects

Whereas direct DNA damage produces fixed positional effects, UVA-induced ROS can diffuse and generate near-neighbor stress. The most abundant and therefore likely primary targets of photons in human skin are chromophores associated with skin structural proteins such as keratin, collagen and elastin. This excitation of dermal extracellular matrix (ECM) protein chromophores that diffuse in the dermis and basal epidermis is referred to as the “bystander model” [16]. Moreover, a diffusion of ROS is observed between the various skin cell types. Indeed, using a two-chamber model, Redmond et al. showed that UVA-exposed cells induced intercellular oxidative signaling to non-exposed cells (keratinocytes, melanocytes and fibroblasts). Melanocytes seemed to be more susceptible to bystander oxidative signaling than keratinocytes and fibroblasts [35].

### 2.2. Biological Effects of Oxidative Stress

#### 2.2.1. Lipid Peroxidation

Lipid peroxidation is a process under which free radicals attack lipids containing carbon-carbon double bond(s), especially polyunsaturated fatty acids, resulting in the formation of lipid hydroperoxides and secondary products including a wide range of aldehyde compounds.

Sebum, which consists of squalene, triglycerides, free fatty acids, cholesterols and cholesterol derivatives including unsaturated covalent bonds in their molecular structure as well as phospholipids and cholesterol present in cellular membranes, can be subjected to lipid peroxidation.

For example, the free radical-catalyzed peroxidation of arachidonic acid results in the production of 8-isoprostane, representing a biomarker of lipid peroxidation. It was shown in reconstructed human skin in vitro that UVA1 exposure induced an increase in 8-isoprostane in a culture medium [33]. UVA1-induced lipid peroxidation was also evidenced in dermal fibroblasts in vitro and in vivo in the dermis of hairless mice by measuring malondialdehyde (MDA) levels [36].

Labile iron directly released in cells by UVA radiation, besides enhancing hydroxyl radical generation, also catalyzes the lipid peroxidation chain reaction [28]. The resulting 4-hydroxynonenal ceramides and oxidized phospholipids are potent signaling molecules modulating the expression of many genes.

#### 2.2.2. Protein Carbonylation

ROS and other reactive substances lead to the oxidative modifications of proteins in the form of carbonylation, i.e., the addition of carbonyl (carbon monoxide, CO) residues into proteins [37]. Protein carbonylation occurs early after UV exposure, either through the oxidative cleavage of proteins, through the direct oxidation of lysine, arginine, proline and threonine residues [38] or by the reaction with reactive carbonyl derivatives [37]. In particular, the peroxidation of lipids in sebum by ROS generates reactive aldehyde compounds (RACs) such as acrolein, which react with amino residues in skin proteins to form carbonylated proteins (CPs) [39].

CPs accumulate specifically within the stratum corneum and in the extracellular matrix of the dermis of photoaged skin, with increased levels in sun-exposed sites compared with sun-protected sites. After UVA exposure, the CPs level increased in the stratum corneum. They could act as a photosensitizer and be a source of ROS. Moreover, their accumulation in the stratum corneum is linked to the loss of skin moisture functions [40].

Interestingly, epidermal layers are less affected by protein oxidation than dermal layers in vivo [41], which is probably due to the far greater antioxidant capacity of the epidermal cells [42]. Protein carbonylation in chronically UV-exposed skin could contribute to the accumulation of structurally and functionally impaired proteins such as elastin, collagen and matrix metalloproteinases [41,43] and be responsible for alterations of the dermal matrix [39]. For instance, repeated exposure to UVA among hairless mice induced the generation of α, β-unsaturated carbonyl compounds such as acrolein or 4-hydroxynonenal (4-HNE), resulting in the formation of adducts on elastin [44]. CPs also lead to changes in the mRNA expression levels of dermal matrix-related proteins such as upregulating MMP-1 and IL-8 and can cause changes in fibroblast morphology. Importantly, in contrast with the dynamic cellular environment, structural extracellular matrix proteins have half-lives lasting decades and are thus highly susceptible to accumulating damage. Taking all these data into consideration, CPs could be involved in the acceleration of the skin aging process [39].

#### 2.2.3. Nuclear and Mitochondrial DNA Damage

##### Nuclear DNA Lesions

The amount of DNA damage and the chemical nature of the lesions depend on the wavelength of the UV radiation. Although the maximal absorption of DNA occurs at 260 nm, corresponding to the UVC range, the absorption of DNA remains significant in the UVB range, with values of 20% for 290 nm and 3% for 300 nm. UVB leads to the formation of cyclobutane pyrimidine dimers (CPDs) and pyrimidine (6-4) pyrimidone photoproducts (6-4 PPs). Interestingly, UV-induced 6-4 PP lesions primarily trigger an apoptotic program in the cell, thus decreasing their carcinogenic potential, whereas CPD lesions appear to principally induce cell cycle arrest [45].

UVA can also induce the formation of CPDs, as well as a large range of oxidatively generated lesions, such as single strand breaks and oxidized bases [46].

Indeed, despite a much weaker direct absorption by DNA, UVA may also induce the formation of CPDs either by direct excitation or via photosensitization, i.e., the absorption of the UV energy by other molecules with the subsequent transfer to DNA that thus indirectly reaches an excited state [47]. The most common mechanism of this photosensitization pathway is called triplet-triplet energy transfer (TTET) [48].

Noticeably, the CPDs generated through this indirect process have been shown to occur after rather than during UV exposure and are thus referred to as “dark CPDs” [49]. In a murine model, UVA-irradiated melanocytes were found to accumulate CPDs after irradiation, with a CPD-level peak detected approximately two hours after UVA exposure before decreasing due to DNA repair. These dark CPDs constitute the majority of CPDs in cultured human and murine melanocytes and in mouse skin, and they are most prominent in skin containing pheomelanin, the melanin responsible for blond and red hair [50]. Although at comparable erythemal doses, UVA1 is less potent than UVB for CPDs formation, the exposure of human skin to UVA1, using a 3D skin model or in human volunteers, revealed CPDs-positive keratinocytes [33,51]. In these studies, the keratinocytes with the highest levels of CPDs after UVA1 were located in the basal layer of the epidermis, corresponding to the proliferative cells at the origin of carcinomas. This location is different from that induced after UVB, where CPDs decreased from the surface to the depth. Another difference was that UVA1, in contrast to UVB, did not induce 6,4 PPs [51].

The direct excitation of DNA after UVA1 irradiation can also occur and is characterized by a specific sequence-context preference for CPD formation, as shown by in vivo studies in mice [52].

Finally, UVA-induced photosensitization also induces a wide variety of oxidative DNA lesions such as single strand breaks and oxidized bases. Among those, the most frequent is 8-oxo-7,8-dihydroguanine (8-oxoGua), which can lead to G-to-T transversions [53,54]; the 8-oxoGua adduct is thus a commonly used DNA marker of UVA oxidative damage [55].

##### Mitochondrial Deletions

Oxidative stress due to UVA1 radiation is also responsible for the generation of large-scale deletions of mitochondrial DNA in human cells in vitro [56] as well as in vivo [57,58].

Indeed, mitochondrial DNA (mtDNA), a circular molecule comprising 16,569 bp in human cells, is particularly vulnerable to oxidative stress due to its lack of histones and limited DNA repair capacity [59]. Accordingly, increased oxidative stress is correlated with an altered mitochondrial function in vivo [60]. Interestingly, the 4977 bp deletion, called the ‘common deletion’, is exclusively present in the dermal compartment of human photoaged skin, indicating that UVA1 plays a major part in the induction of mitochondrial DNA mutations in human skin [61].

Furthermore, the induction of the common deletion is proportional to the number of UVA1 exposures given to cells, showing that it results from cumulative photodamage [56]. Sun-induced mitochondrial DNA damage would be an important factor in the pathogenesis of photoaging (see Section 2.5.2).

#### 2.2.4. Apoptosis

UVA1-induced apoptosis has been widely studied with regard to its therapeutic advantage leading to T-cell apoptosis (see Section 3.1). UVA1 is able to elicit two different apoptotic pathways [62,63]: it causes singlet-oxygen damage that depolarizes mitochondrial membranes, triggering immediate apoptosis (T ≤ 4 h), and also oxidative damage to DNA, inducing delayed apoptosis (T ≥ 24 h). By contrast, UVB only causes delayed apoptosis [62]. The ability of UVA1 to induce a combination of different apoptotic cell death mechanisms may preclude the appearance of resistant mutant cell populations [63]. Other dermal cell types such as endothelial cells have also been reported to be targeted [64].

However, UVA1 exposure in vivo did not induce apoptosis in the epidermal cells [64]. These results were confirmed in reconstructed skin exposed to UVA1 radiation, in which a major apoptosis was observed in superficial dermal fibroblasts but not in epidermal keratinocytes [33,65]. In line with the disappearance of dermal fibroblasts, genes encoding proteins related to apoptosis (DDIT3, NR4A1 and IER3) were upregulated at earlier time points after UVA1 exposure [33].

#### 2.2.5. Alteration of Energy Metabolism

Several studies evidenced a disturbance in cellular energy production following exposure to UVA1.

First, as previously mentioned, UVA1 induces mtDNA mutagenesis. This was associated with a decline in mitochondrial functions, i.e., in oxygen consumption, mitochondrial membrane potential and ATP content [66], reflecting a perturbation of energy metabolism. In line with these data, it has been shown that the exposure of HaCat keratinocytes to UVA leads to a reduction in ATP production [67].

More recently, energy metabolism changes were also evidenced after UVA1 exposure, using an in vitro 3D skin model and the non-invasive imaging of nicotinamide adenine dinucleotide (NAD(P)H) and flavine adenine dinucleotide (FAD) by multicolor two-photon fluorescence lifetime microscopy [68].

Interestingly, the UVA1 exposure of in vitro reconstructed skin led to the modulation of the expression of genes related to glucose metabolism. Three pyruvate dehydrogenase kinase genes (PDK1, PDK2, PDK3), the ALDOC gene encoding a glycolytic enzyme and the H6PD gene (hexose-6-phosphate dehydrogenase) were down regulated, while PGM3, PYGB and UGDH were upregulated, attesting a disruption in glycolysis and glycogen degradation [33].

The alteration of energy metabolism may have a role in immunosuppression (see Section 2.4.2).

### 2.3. From UVA-Induced DNA Mutations to Skin Cancer

UV radiation induces DNA lesions that can lead to mutations in proto-oncogenes and tumor suppressor genes such as p53 [69,70]. In line with this, the whole UV spectrum has been shown to trigger p53 accumulation, although there are some differences in the p53 location throughout the epidermis after UVB and, preferentially, in the basal layer after UVA [71,72]. Studies performed with repetitive exposures to low (even very low) doses of UVA1 confirmed the p53 basal accumulation [73,74].

In the absence of an efficient repair of DNA damage before replication, mutagenesis and then carcinogenesis can occur.

UV radiation has been linked to the three major types of skin cancer: basal cell carcinoma (BCC) and squamous cell carcinoma (SCC), which are nonmelanoma skin cancers (NMSC), and malignant melanomas (MM) [75]. Di-pyrimidine mutations are commonly found in UVR-induced carcinomas, and their direct causal role has been demonstrated long ago [76,77,78,79,80]. According to a mouse model and after correction for differences in epidermal UV-transmission between mouse and human skin, the predicted contribution of UVA to the induction of NMSC, which are ten times more frequent than MM, appears substantial (10–20%) in humans [81]. Mutations in sunlight-induced melanomas [80] also arise from CPDs, showing that the main damaging process involves the dimerization of pyrimidine bases through the direct absorption of UVB and, to a lesser extent, of UVA photons [45,76,77]. However, although fewer dimers are formed with UVA, a less efficient cell cycle arrest with UVA may make UVA-induced pyrimidine dimers more mutagenic than UVB-induced ones [82].

While UVB-induced mutations are mostly G:C->A:T at dipyrimidiques sites and tandem CC->TT (or GG:AA) transitions, UVA-induced mutations are rather G:C->T:A (a hallmark of the 8oxoG lesion) and A:T->C:G transversions, as well as G:C->A:T transitions at non-dipyrimidique sites and at TCG sites (resulting from CPD lesions) [83,84,85,86].

In human skin, few studies have been performed on UVA-induced mutations. p53 mutations were analyzed in the keratinocytes of human skin exposed to UVA1, revealing G-to-T transversions [87]. In reconstructed human skin exposed to UVA, mutations were induced in the p53 gene (mostly A:T-> C:G and G:C->C:G transversions) [88]. Importantly, UVA1-induced mutations derived from 8-oxoGua and CPD lesions are found in the basal epithelial layers and could lead to fixed genomic mutations if left unrepaired [4,32,79,80]. In particular, only low levels of the specific DNA repair enzyme 8-oxoguanine-DNA glycosylase 1 (OGG1) have been detected in basal cells [8]. Since basally located stem cells give rise to skin tumors [89], UVA appears as an important carcinogen in the stem cell compartment of the skin. Its early involvement in carcinogenesis is illustrated by the identification of UVA-induced DNA mutations in transformed keratinocytes from both SCCs and premalignant solar keratosis in almost identical proportions [90].

Finally, in a clinical study including 62 subjects from European, Asian or African descent, the expression of genes involved in the regulation of cell proliferation and differentiation was shown to be modulated six hours after UVA1 exposure, even in individuals with highly pigmented skin [13]. Two genes in particular were regulated in a similar way in the three populations. The first one, the adenomatosis polyposis coli downregulated 1 (APCDD1) gene, known to regulate skin development, homeostasis, pigmentation and cancer as an inhibitor of the Wnt signaling pathway [91], was downregulated. The second, the ornithine decarboxylase 1 (ODC1) gene, coding for the first enzyme in polyamine synthesis and essential in the normal growth, differentiation, and invasiveness of keratinocytes during tumorigenesis promotion [92], was upregulated.

Studies also support the role of UVA in the development of UV-induced melanoma. It has been shown in a transgenic mouse model of UV-induced melanoma that UVA exposures play a critical role through the formation of oxidative DNA damage in melanocytes [93]. Importantly, the same team also clearly established a link between UV exposures in the neonatal period and susceptibility to melanoma induction [94]. Of great interest, the dose of UVA used in Noonan et al.’s 2012 study [93] corresponded to 6–12 min of tanning bed use, supporting the involvement of UVA-emitting tanning devices in melanoma development [95].

The mutagenic property of UVA thus makes it a larger contributor to UV-induced skin carcinogenesis in humans than previously thought, especially as it is combined with UVA1-induced immunosuppression (see next paragraph) [8].

### 2.4. Impact on Immunity

#### 2.4.1. Inflammatory Signalization and Inflammation/Erythema

Erythema is a manifestation of solar-induced inflammation which can also be produced by UVA, although receiving a standard erythema dose (SED) requires a four-times-longer exposure of sunlight to the UVA component alone than the full UVR spectrum in an individual light phototype (I-II) during mid-summer in Chilton (UK) [4].

UVA1 triggers several cellular inflammatory mechanisms, especially through cytokine alterations mediated by oxidative pathways, which have been demonstrated in several studies [96,97,98,99].

UVA1 exposure was shown to induce the proinflammatory cytokines tumor necrosis factor (TNF)-α and interleukin (IL)-8 expression in the human epidermoid carcinoma cell line KB at the mRNA and protein levels [96]. Moreover, a synergistic effect of UVA1 was observed on UVB-induced release by human keratinocytes of IL-8 [97], a potent proinflammatory chemokine which acts as a chemotactic and activation factor for neutrophils and T lymphocytes [100] and induces the proliferation of human keratinocytes in vitro [101]. Neutrophils may also be important in the regulation of inflammatory and immune responses of the UV radiation-induced skin inflammation [99].

In a reconstructed human skin model, among the genes modulated after UVA1 exposure, approximately 14% and 10% were related to inflammation in keratinocytes and fibroblasts, respectively, with genes strongly upregulated as early as 2 h after UVA1 exposure. In particular, pro-inflammatory markers such as IL-6 and C-C Motif Chemokine Ligand 20 (CCL20), which are important in mediating leukocyte recruitment, and colony-stimulating factor 2 (CSF2) genes and proteins, were strikingly upregulated [33].

In another study in human volunteers, six hours of exposure to UVA1 led to the upregulation of prostaglandin endoperoxide synthase 2 (PTGS2) and IL-8, suggesting a proinflammatory response [13]. Importantly, a significant increase in IL-8 was measured in individuals from three different ethnic backgrounds, even in the most highly pigmented volunteers, demonstrating that inflammation is triggered by UVA1 irradiation, irrespective of the phototype.

The UVA-induced generation of ROS, either directly and or via the generation of lipid peroxides, also induces intercellular adhesion molecule-1 (ICAM-1) mRNA and surface expression. ICAM-1 plays a pivotal role in the generation and maintenance of immunologic inflammatory skin diseases by mediating leukocyte–keratinocyte adhesion [102]. The upregulation of ICAM1 by UVA1 exposure may be relevant in the pathogenesis of polymorphic light eruption [103]. In keratinocytes in vitro, the UVA-induced transcriptional activation of ICAM-1 is mediated by the activation of the transcription factor AP-2 [104].

#### 2.4.2. Immunomodulation

##### In Vivo Immunosuppression

In vivo, the suppression of the skin immune function can occur long before erythema with 25–50% of the minimal erythema dose (MED) in subjects with skin types I/II [105].

The immunosuppressive effect of UVA1 is supported by the suppression of the delayed-type hypersensitivity (DTH) response to recall antigens. Studies under real sun-repeated exposure, UVA, UVA1 or using various levels of UVA and UVB protection unambiguously demonstrated that UVA rays are particularly involved in photoimmunosuppression [106,107,108]. The precise contribution of the different UV wavelength domains has been studied more recently in a nickel model of recall contact hypersensitivity and revealed that solar-induced immunosuppression peaks at 300 nm (UVB) and 370 nm (UVA1) (Figure 3). UVA1 rays, which are present in the majority in sunlight, lead to a threefold greater immunosuppression than UVB rays at low doses, which can be received during normal daily activities [12], whereas UVA2 has not been shown to be immunosuppressive in humans [109].

By contrast, with the linear dose responses of UVB-induced immunosuppression, UVA1 produces bell-shaped, Gaussian dose responses, suggesting different chromophores and mechanisms of immunosuppression for UVA [109]. Two hypotheses can account for this bell-shaped dose-response curve: (i) medium doses of UVA1 radiation could initiate immunosuppressive mechanisms, such as ROS production, while greater doses could trigger antioxidant mechanisms; (ii) greater doses could inactivate a critical component of the immune suppressive pathway, such as a photolabile chromophore [8].

##### UVA1 Cellular Impact on Immune Cells and Signaling

○UVA1-induced immunomodulation via effects on Langerhans cells

UVA affects Langerhans cells (LC) in several ways, which may lead to an alteration of the skin immune response. Although LCs account for about 2% of the epidermal cell population, their long dendrites enable them to sense most events occurring in the whole epidermis [110].

UVA was shown to induce alterations of the density and⁄or morphology of LCs. Indeed, Dumay et al. reported fewer epidermal LCs in UVA1-irradiated human skin 3 days post-exposure, as well as a significant reduction of the class-II major histocompatibility complex (MHC II) antigen HLA-DR expression on viable epidermal LCs [111]. Regarding LC morphology, a rounding up of the cell body with a decrease in the length of LCs and a reduction of dendrite size was observed by electron microscopy after total UVA or UVA1 exposure, especially for the highest doses (50 J/cm^2^) [112]. As a result, the cell contacts between LCs and keratinocytes appeared to be loosened. In addition, alterations were also observed in LC organelles. They included noticeable mitochondrial alterations, such as swelling, the partial-to-total dissolution of the inner membranes and the dilation of the reticulo-endothelium and Golgi apparatus at the highest dose of either UVA1 or total UVA radiation. These damages to the organelles may be triggered by the UVA-induced generation of ROS [112]. Altogether, the alteration of mitochondrial function and an excess of ROS may lead to LCs apoptosis [112].

By contrast, UVA1 did not seem to directly interfere with the antigen-presenting cell function of LCs, unlike UVB [113,114].

##### UVA1 Impact on T Lymphocytes

As shown in vitro, UVA1 can kill transformed T lymphocytes by inducing apoptosis [115], particularly cytotoxic T lymphocytes with marked CD8 positivity [116]. In mice, in vivo, after contact sensitization, UVA does not affect the number of CD8+ T lymphocytes but rather impairs their development into long-term memory cells [117].

In humans, in vivo, taking advantage of the depth of penetration combined with the immunomodulating properties of UVA1 rays, UVA1 phototherapy is used to treat several immune-mediated skin diseases, such as atopic dermatitis, lupus erythematosus or morphea. Due to the importance of T lymphocyte subsets in these pathologic conditions, their levels were studied in patients treated by UVA1 phototherapy. In morphea or in atopic eczema patients, it appeared that regulatory T cells (T-reg cells) were not particularly affected by UVA1 exposure [118,119]. In contrast, UVA1 phototherapy decreased the T helper 1 and T helper 2 Th1/Th2 balance as well as the T cytotoxic Tc1/Tc2 ratio in patients with systemic lupus erythematosus. This decreased proportion of IFN-gamma-secreting cells could explain the beneficial effects of UVA1 phototherapy in systemic lupus erythematosus [120].

##### UVA-Induced Immunosuppression via Urocanic Acid

Urocanic acid (UCA) is a skin chromophore which plays an important role in the mechanism of photo-induced immunosuppression [121,122,123]. The naturally occurring trans isomer is a major natural moisturizing factor and maintains the skin’s acid pH [123]. It is produced in the skin by an enzymatic deamination of the amino acid histidine and accumulates in the stratum corneum. Following UV exposure, UCA is isomerized to the cis form, which has been shown to induce immunosuppression when applied to normal skin. In human volunteers, acute or repeated exposure to UVA or to SSR induces a dose-dependent increase of the cis isomer of UCA [124]. Moreover, the formation of cis-UCA in the skin involves a broad range of UV wavelengths, up to at least 363 nm, which shows the ability of UVA to isomerize UCA [125].

##### UVA1-Induced Immunomodulation via Calcineurin

UVA1 irradiation may suppress the skin immune system by inactivating calcineurin (Cn) in the skin. Calcineurin is a eukaryotic Ca^2+^- and calmodulin-dependent serine/threonine protein phosphatase which plays an important role in T cell activation, controls the synthesis of IL-2, γ-interferon, IL-4 and IL-10 and constitutes a target for immunosuppressant drugs. In the Jurkat T-lymphoma cell line and peripheral blood mononuclear cells (PBMC), UVA1 reduced Cn activity, as well as the production of IL-2, γ-interferon, IL-4 and IL-10 [126]. Additional in vitro experiments suggested a potential direct in vivo effect of ROS generated by photosensitization on Cn activity following UVA1 exposure [127,128].

##### UVA1-Induced Immunomodulation by Other Mechanisms

UVA1 exposure changes the NADH and FAD lifetimes and reduces the ATP content in keratinocytes [67,68]. Such mechanism may contribute to the immunosuppressive action of UVA in vivo, since vitamins essential for energy metabolism, such as nicotinamide or riboflavin, prevent UVA1 (385 nm)-induced immunosuppression in humans [8,129,130].

Other mechanisms of UVA-induced immunosuppression involve the activation of the alternative complement pathway, which may act as a sensor of UVA-induced damage in the skin [8].

Besides these studies, Marionnet et al. observed in UVA1-exposed reconstructed human skin a downregulation of genes involved in antiviral and antibacterial defense, such as interferon inducible genes (SAMD9, SAMD9L, IFIT1, IFIT2, IFIT3, MX1, MX2, OAS1, OAS2, GBP2, GBP5, GBP6, CXCL10), genes encoding receptors to double-stranded RNA (TLR3, DDX58) and C-type lectin receptors (CLEC2A, CLEC2B) [33,131]. In addition, UVB was shown to downregulate IFN signaling, which could contribute to UV-induced immunosuppression [132,133]. The downregulation of the interferon response by UVA1 could then contribute to immunosuppression and to the reactivation of the herpes simplex virus [134,135].

Besides photoimmunosuppression, UV’s impact on immunity can also translate into sun-induced allergies.

##### Role of UVA1 in Photodermatoses

Polymorphic light eruption (PMLE) and solar urticaria (SU) are two photodermatoses that are induced by UV radiation and sometimes by Vis [136].

PMLE is the most common endogenous photodermatosis, affecting mostly women but also men (~25% of cases) of all ages. It commonly occurs in individuals with Fitzpatrick skin types I–IV [137] one or two days after the exposure to sunlight, depending on patient sensitivity and on exposure conditions. It is characterized by erythematous papules, plaques and vesicles that are often accompanied with pruritus and appear only on the sun-exposed areas of the skin. Patients with PMLE, which is considered as an autoimmune-mediated skin condition, exhibit the failure of normal UV-induced immunosuppression, probably leading to diminished self-tolerance and enhanced reactivity to photoallergens [138].

The solar wavelengths involved in triggering PMLE may include the entire UV spectrum, but it is commonly held that PMLE is mainly caused by UVA rays (UVA1 or UVA1 + 2) [139] and can be experimentally induced by UVA1 [140]. As reported by H. Honigsmann [141], UVA has been more effective than UVB in inducing PMLE. In two studies, following exposures to UVA or UVB, the action spectrum was predominantly in the UVA range (56% for UVA, 17% for UVB and 26% for both UVA and UVB [142]; 68% for UVA, 8% for UVB and 10% for both wavelengths [143]).

The variation in the proportions of UVA and UVB present in terrestrial sunlight may also explain the higher incidence of PMLE in temperate rather than tropical regions, with greater susceptibility to the condition in spring and, occasionally, autumn [137,144].

In line with the development of PMLE under cumulative moderate UV exposures, an indoor study carried out on 52 women of Caucasian origin (Fitzpatrick phototypes II–IV) who were prone to developing this disorder showed that relatively low UVA doses (a cumulative UVA dose of 80 J/cm^2^ over a 3-day period) are able to induce PMLE lesions [139].

In contrast with PMLE, idiopathic solar urticaria (SU) is uncommon and manifests within minutes of sun exposure. The published literature consists mainly of single case reports and, recently, a small number of cohort studies (for a review, see [145]). Solar urticaria is a mast cell-mediated disease characterized by the early onset of itching and erythematous wheals after exposure to UVA and Vis radiation and, less commonly, UVB and IR.

### 2.5. Links between Dermal Damage and Photoaging

It has been reported that approximately 80% of skin aging on the face of individuals of Caucasian descent can be attributed to chronic sun exposure [146]. Both UVB and UVA contributed to photoaging, as shown by the analysis of photoaging markers in chronically exposed mice [147,148,149]. UVA1 is now established as a major promotor of skin aging [150] and is also responsible for skin “sagging” [51].

Photoaging is characterized by relatively mild structural epidermal changes [151] but major modifications in the dermis involving, in particular, ECM [152]. Indeed, the dermis, which is the nonproliferating compartment of the skin, would be as important, if not more so, than the epidermis in initiating and sustaining the process of extrinsic skin aging [153]. The penetration properties of UVA and, particularly, UVA1 strongly support their involvement in the photoaging process. Indeed, individuals exposed to UV through glass and who therefore receive the longest UVA rays on one side of the face, such as truck drivers, present aggravated signs of skin photoaging on the exposed side compared to the other. The role of UVA in such unilateral dermatoheliosis development, with drastic dermal cutaneous alterations and the formation of deep wrinkles, has clearly been emphasized [154].

In the dermis, both extracellular and cellular alterations have been described.

#### 2.5.1. Solar Elastosis

Solar elastosis, one of the main dermal characteristics of photoaged skin, consists of an accumulation of disorganized and non-functional degraded elastic fibers that appear dystrophic, fragmented and thick. This accumulation of degenerative elastotic material includes elastin, fibronectin, fibrillin, abnormally localized and organized microfibril-associated proteins (MAGP-1, -4), glycosaminoglycans, lysozyme and matrix metalloproteinases such as MMP7 (matrilysin), MMP12 (elastase) and MMP-1,-2,-3 and exhibits the degradation of collagen fibers as well as changes in the ratio of the different types of collagen [155,156,157,158]. Alterations of elastic tissue are supported by the induction of MMP-12 (human macrophage metalloelastase) [159], the skin fibroblast-derived elastase [160] and neutrophil elastase [161]. Other components of the ECM homeostasis are modified. Among the glycosaminoglycans (GAGs), dermal hyaluronic acid (HA) is significantly increased in photoaged skin together with a decrease in the expression of hyaluronan synthase 1, an increase in hyaluronidases 1–3 and a decrease in the HA receptors CD44 and receptor for HA-mediated motility (RHAMM) [162]. Moreover, the carbonylation of dermal proteins, likely to be linked with the high susceptibility of dermal tissue to protein oxidation with repeated UV exposure, could also contribute to solar elastosis and premature aging [41]. The balance between the synthesis and degradation of ECM proteins is maintained by MMPs. They play a key role in the elimination of damaged or oxidized proteins and in tissue remodeling. However, their chronic activation and excessive levels in the dermis, together with sustained oxidative protein damage, lead to irreversible damage to the ECM, which can impair the structural integrity of the dermis and progressively erode the properties of the skin. This progressive impairment of dermal structure organization has been modelized by Fisher et al., assimilating solar elastosis formation to an abnormal repair process and leading to a “dermal solar scar” [163].

#### 2.5.2. Contribution of UVA1 to Photoaging-Related Dermal Alterations

In the UV spectrum, due to their penetration properties, UVA1 rays can directly target the dermis.

The role of UVA up to 400 nm in dermal alterations has been established a long time ago [164,165]. UVA1 rays alone have been shown to be as effective as total UVA, indicating the key contribution of the longest wavelengths [166].

UVA1 can trigger two of the major pathways leading to an impaired balance between the ECM synthesis/organization and degradation involved in photoaging [61]: the decrease or alteration in the major structural proteins of dermal ECM composition and the induction of MMPs leading to ECM breakage and fragmentation.

First, UVA1 induced the modulation of the expression of genes encoding proteins involved in ECM composition. As shown in dermal fibroblasts or in reconstructed skin, Collagen Type I Alpha 1 Chain (COL1A1) gene expression is downregulated after UVA1 exposure [33,167] (Figure 4). In vivo, the data confirmed that the UVA1 exposition of 14 European females for 13 weeks enhanced the expression of tenascin and increased lysozyme and alpha-l antitrypsin deposition on elastic fibers in the dermis similarly to what can be observed in solar elastosis [168]. This UVA1 effect was also found to be at a higher extent than that of solar simulated radiation (total UV spectrum) at equal suberythemal doses.

Secondly, UVA1 exposure can stimulate the ECM degrading process through the increase in MMPs. This effect has been shown in a 2D fibroblasts cultures as well as a 3D reconstructed skin model with the upregulation of MMP1 and MMP3 genes or protein expression [33,169] (Figure 4).

The modulation of MMPs expression in human skin has also been shown in vivo after UVA1 exposure with the upregulation of MMP1, MMP3 and MMP10 mRNA 24 h after exposure [170]. In parallel, the expression of MMP12 (mRNA and protein) was specifically induced by UVA1 [170]. MMP12 is the major enzyme for the degradation of elastin [171] and is likely to contribute to the loss of elasticity in late solar elastosis. UVA1 would induce MMP12 via ROS generation [159], especially singlet oxygen. The upregulation of MMP-1 was also confirmed in recent in vivo studies after UVA1 exposure in lightly to moderately pigmented individuals but also in Indian subjects with a more pigmented skin color type [10,13]. No MMP-1 gene modulation was detected in African subjects, which could correlate to the less severe photoaging observed in heavily pigmented skins [13].

Moreover, chronic and sub-erythemal inflammation is a main contributor of the aging process [172]. This low-grade inflammation, which can initiate or significantly influence the changes in skin aging signs, is sometimes referred to as “inflamm-aging” [173,174]. For example, UVA has been shown to induce a surge of IL-6 in interstitial skin fluid, which in turn stimulates fibroblast-derived MMPs [98,175].

#### 2.5.3. Contribution of Mitochondrial Dysfunction in Dermal Photoaging

Several independent studies have demonstrated that large-scale deletions of mtDNA are increased by up to 10-fold in photoaged skin compared with sun-protected skin from the same individual [176,177]. They are mostly located in the dermal compartment and are linked to the repetitive exposure to physiological doses of UVA1 [57]. Mitochondrial DNA damage is thus considered as a relevant biomarker for the UV exposure of skin [178]. According to Krutmann et al. [179], the accumulation of sun-induced mitochondrial damage in skin fibroblasts represents a key factor in the pathogenesis of the photoaging of human skin. Indeed, it leads to an altered gene expression in fibroblasts, which promotes both the dermal (e.g., wrinkle formation) and epidermal (atrophy, barrier dysfunction) aging process [180]. In vitro, the UVA-induced generation of the common deletion in skin fibroblasts was associated not only with a decrease in mitochondrial function but also with an upregulation of the collagen-degrading enzyme MMP-1 [181]. To further support the role of mtDNA deletion in the development of clinical signs of skin aging, a recent publication showed that, by using an inducible mtDNA depleter mouse model, there is a correlation between mtDNA deletion and the development of skin wrinkles [182]. They also confirmed the association with the upregulation of MMPs and a pro-inflammatory response. More importantly, the clinical aging signs, such as wrinkles, can be reverted by turning off the transgene expression, leading to a restoration of mitochondrial functions.

### 2.6. UVA1-Induced Pigmentation and Pigmentary Disorders

#### 2.6.1. Immediate and Persistent Pigment Darkening and Delayed Tanning

Under sun exposure, hyperpigmentation leads to a visible darkening of the skin as a result of the impact of the different wavelength domains and involves several mechanisms and time courses. The three main phases of UV-induced pigmentation comprise the immediate pigment darkening (IPD), the persistent pigment darkening (PPD) and the delayed tanning (DT) [183].

The most described UVA1 short-term consequence, which peaks between 340 and 370 nm, is IPD-characterized by an immediate gray-brown discoloration or pigmentation. IPD fades rapidly, but it can be followed by a residual pigmentation—PPD—after UVA doses > 10 J/cm^2^ [13,169,170]. The UVA-induced pigmentation, in contrast with UVB-induced tanning, mostly involves a pigment darkening response rather than melanogenesis activation, and it does not protect against DNA photoproducts formation [184,185]. Indeed, UVA-induced IDP/PPD results from the photooxidation of preexisting melanin or melanin precursors and their metabolites and/or the polymerization of melanin precursors and/or some redistribution of pigment granules, whereas UVB mostly upregulates melanin synthesis, leading to DT [184]. PPD corresponds to a brown pigmentation that lasts at least 24 h or longer [186]. Noticeably, UVA1 rays were shown to be the main contributors to skin darkening, producing irreversible brownish black pigmentation [186]. Dark-colored pigment generated outside of melanocytes in response to UVA exposure, by the photooxidation of metabolites of DHICA (5,6-dihydroxyindole-2-carboxylic acid) in the basal and suprabasal layers of the epidermis, is likely involved in skin-persistent pigmentation without reddening observed after solar exposure [186]. PPD would develop with an irreversible chemical process including the oxidative cleavage of indolequinone [187].

PPD may blend with the DT response linked to melanogenesis activation in the following days [188,189]. Recently, a molecular mechanism for UVA-induced melanin, involving a G protein-coupled receptor, a calcium flux increase and the generation of mitochondrial ROS, was described in melanocytes in vitro [190]. Another potential mechanism for UVA1-induced melanogenesis, especially in repetitive exposure conditions, may involve DNA damage and subsequent p53 accumulation [191].

Importantly, a single UVA1 exposure can induce both immediate and long-lasting skin darkening with a similar amplitude in skin phototypes III to VI (Figure 5) [13]. In highly pigmented skins, it can lead to a more pronounced grayish aspect, which is noticeable for these populations who want to avoid skin darkening and the development or exacerbation of hyperpigmented lesions [192].

UVA1-induced darkening of the skin contributes to no photoprotection while inducing DNA and cellular damages [184]. Moreover, pigmented skins up to phototype VI, usually considered to be well protected against UV-induced damage, also show that UVA1-induced hyperpigmentation is associated with changes in the expression of genes involved in different biological functions such as inflammation or oxidative stress responses, thus demonstrating a real cellular impact [13].

#### 2.6.2. Pigmentary Disorders

Cutaneous hyperpigmentations represent a group of diseases linked to hypermelanosis [194]. It is estimated that these disorders of pigmentation motivate dermatological consultation, with approximately 24.7 million dermatology visits made between 1994 and 2010 for the management of dyschromia [195,196,197]. In a recent study, discoloration/abnormal pigmentation concerns were found as the third reason for dermatology visits in the US [198].

Many of the hyperpigmented disorders are influenced by sun exposure due to its pro-melanogenic role. The most frequent ones include melasma, post-inflammatory hyperpigmentation and solar/actinic lentigines.

Actinic lentigines are considered as photoaging lesions only observed on chronically photoexposed areas [199]. Melasma has been more recently considered as a UV-related photoaging pathology [200,201]. For post-inflammatory hyperpigmentation (PIH), which occurs after inflammatory skin diseases or external injuries such as laser treatment or surgery, the role of ambient daily light exposure has been demonstrated to be the triggering factor [202].

Hyperpigmentation of the skin stands for one of the main dermatological concerns for populations with pigmented skin phototypes [192] and is highly prevalent in Asia [203] and India, where high annual doses of UVA1 radiation are received [204]. A large study in 1204 women from four Indian cities showed skin color heterogeneity in more than 80% of them, independently of age. It mainly resulted from hyperpigmented spots, melasma and ill-defined patchy pigmented macules and dark circles [205].

Similar high incidences for pigmentary disorders can also be observed in dark-skinned individuals from other descents in other regions of the world [206]. Indeed, as melanocompetent individuals have more responsive melanocytes, they are at a greater risk of developing more prominent and longer lasting hyperpigmentation than individuals with lighter skin [196].

The contribution of UVA1 is now clearly taken into account, as highlighted in the recent photoprotection recommendations [207] (see Section 5).

## 3. Lessons Learned from UVA1 Phototherapy and the Use of Tanning Beds

### 3.1. Phototherapy

Phototherapies such as photodynamic therapy (PDT), UVA1, UVB and psoralen-UVA (PUVA) photochemotherapies are used to treat dermatological diseases. These phototherapies function by altering cytokine profiles, changing the skin immune cytotoxicity, and directly killing diseased cells by apoptosis. UVA1 phototherapy in particular was first reported as a skin disease treatment in 1981 [208] and was developed in the last 20 years. It shows a comparable efficacy to other types of phototherapy, such as narrowband UVB, for some skin conditions [209]. UVA1 phototherapy is commonly used in the treatment of immune-mediated skin diseases [210] such as atopic dermatitis, lupus erythematosus and urticaria pigmentosa. In patients with atopic dermatitis, in which the immune response is exacerbated [211], UVA1 therapy used to manage flares was found to be accompanied by a reduction in IL-5, IL-13 and IL-31 mRNA expression, an increase in the expression of IL-8 and a decrease in the serum level of eosinophil cationic protein [212]. At the cellular level, it was linked to reductions in the number of dermal LCs, activated eosinophils and mast cells and, in a relative number of intraepidermal IgE+ LCs, was linked to an increase in the percentage of dermal CD8+ T lymphocytes [213]. These changes were linked to a substantial clinical improvement of the skin by reducing the excessive immune reactivity of the skin.

UVA1 phototherapy is also a common therapeutical approach for fibrosing conditions such as localized pansclerotic morphoea or scleroderma [214]. The efficacy of these treatments is not fully understood but seems to rely on immunomodulation, the induction of collagenases and the initiation of apoptosis.

UVA1 exposure during phototherapy sessions can lead to acute side effects such as hyperpigmentation, skin dryness, erythema, pruritus, herpes simplex virus reactivation and PMLE, the latter being one of the most described [6,122,195,201,202]. Chronic side effects are also mentioned, such as photoaging signs and possible carcinogenesis [215].

Both the mode of therapeutical action of UVA1 phototherapy (decreased immune responses, ECM degradation) and its reported side effects are pieces of evidence supporting the role of such wavelengths in their development in individuals exposed to solar UVA1.

### 3.2. Tanning Beds

Another setting of exposure to artificial UVA1 rays is the use of sunbeds and artificial tanning devices. Their radiation is richer than sunlight in UVA, especially UVA1, and there is growing evidence for their involvement in MM and, to a lesser extent, in SCC [216,217]. Indeed, epidemiological studies showed a strong association between the use of sunbeds and MM incidence, with most of the sunbeds emitting mainly UVA (only 0.1% to 2.1% of the total UV dose derived from UVB) [218,219].

Sunbed use may also contribute to skin photoaging. They induce photoaging-related mtDNA deletions in the dermis, as shown in vivo [220], and also age-related pigmentary problems such as idiopathic guttate hypomelanosis or hyperpigmented lesions. PUVA lentigines and sunbed lentigines have been reported, supporting the fact that UVA exposure is involved in melanocyte dysplasia [221]. An examination with the ultraviolet light-enhanced visualization (ULEV) method of the skin of 33 young women (phototype III) designated as avid users of sunbeds and as involved in frequent exposures to sunshine for a duration of at least 120 months showed that they presented a significantly increased number of pigmentary changes [222]. They corresponded to mottled subclinical melanoderma (MSM), as well as rare spotty amelanotic macules presenting as skin ivory spots (SIS).

Regarding the effect of sunbed use on the physical properties of the skin, it has been shown that repeated exposures to sub-erythemal UVA doses with tanning beds caused sagging skin due to an increase in extensibility and hysteresis and a concomitant decrease in the elasticity of the skin [223]. A more recent study of 65 women aged 31–46 years with frequent exposures to tanning sunbeds and sunshine (phototype III) over a period of 8 years showed that the increase in both skin extensibility and hysteresis and decreased elasticity were progressive [224]. These observations indicate that the continuous use of sunbeds leads to a functional decline in the dermal structure similar to that of premature aging.

Figure 6 sums up all the above described effects from natural to artificial exposure to UVA1 (Section 2 and Section 3). 

## 4. Combined Effects of UVA and Other Environmental Factors

### 4.1. UVA and Pollutants

Among the external factors that are part of the skin exposome, airborne pollution has received great interest in the last decade [225]. Several studies have indeed pointed out the role of air pollution in the aggravation/acceleration of skin aging signs such as wrinkles and hyperpigmented lesions on the face [226,227]. In a recent study, by comparing a similar population from two Chinese cities located at the same latitude (thus submitted to approximately the same sun exposure) but exposed to different atmospheric pollution levels, it was possible to demonstrate that living in a highly polluted environment for many years increased the prevalence of hyper-pigmented lesions (e.g., simplex lentigo) as well as the severity of wrinkles [228]. Importantly, some pollutants, such as polycyclic aromatic hydrocarbon (PAH) moieties are photo-reactive and phototoxic, particularly with UVA [229]. Sunlight and pollution might thus synergistically contribute to skin damage [226].

Although pollutants may interact with the skin surface, the contamination of deep skin by particulate matter (ultrafine particles or some PAH) moving through the bloodstream is also highly likely. Indeed, in polluted areas, PAH were detected in the cord blood of neonates [230] and in the cortex of hair of women [231]. The resulting concentrations of contaminant PAH in the skin are likely to be in the nanomolar range, as previously measured in the blood of smokers [232]. At such concentrations, pollutants (PAH, Particulate Matter or Particulate Matter extract) lead to ROS generation within keratinocytes and, in particular, in the mitochondria, with membrane depolarization and/or reduced ATP production as consequences [233]. In addition, intracellular glutathione concentrations decreased several hours after treatment with PAH and potentiated PAH phototoxicity. In parallel, keratinocytes exposed to the combination of benzo[a]pyrene and indenopyrene at very low concentrations and exposed to UVA1 have impaired proliferation and clonogenic potential [233], as well as hampered DNA repair [234]. In a reconstructed skin model, UVA1 combined with benzo[a]pyrene led to drastic epidermal alterations [235].

This synergy can have effects even with short-term environmental exposure (from minutes to hours and no more than a week in humans), leading to increased tissue peroxidation and decreased cutaneous alpha-tocopherol, causing additive oxidative stress in the stratum corneum [212,213,220].

In conclusion, the effects of pollution appear to constitute an emerging part of the “skin exposome”, substantially adding on to the effects of sunlight, especially the UVA1 wavelengths domain [11,141,213].

### 4.2. UVA1 and Visible Light

Sunlight is a mixture of radiations that could have additive or synergistic effects on skin by giving rise to various and interactive genotoxic modifications and cellular alterations. Indeed, while targeting different chromophores, Vis and UV have all been reported to generate ROS within cells, leading to increased ROS [236]. Although the damaging properties of Vis on skin are not completely assessed, the main described clinical impact of Vis is an increase in skin pigmentation. This effect can be induced by the Vis portion of natural sunlight (400–700 nm) and represents a contributor of sunlight radiation in hyperpigmentation [237]. This can also be observed after exposure to artificial Vis light sources, metal halogen [238] or solar simulation [239].

In humans, Vis can trigger skin type-dependent pigmentary alterations. The pigmentation induced by Vis was shown to be darker and more sustained than that induced by UVA1 in individuals with skin types IV–VI, whereas no pigmentation was observed in skin type II under both types of radiation at doses corresponding to a 1 h sun exposure [238]. Furthermore, the addition of trace amounts of UVA1 (<0.5%) to Vis provided a synergistic effect on skin pigmentation intensity compared to the effects of Vis alone [240]. In these same subjects with skin types IV–VI, clinical erythema was observed with the combination Vis + UVA1 but not with pure Vis. The combination of Vis and UVA1 (2%) also increased erythema immediately after irradiation in subjects with skin types I-III [241].

By using solar simulation, physical filters and a clinical design enabling a conclusion about the relative contribution of particular wavelength ranges, we could show that the pigmentation induced by UVA1 and by HEV (400–450 nm) was cumulative (manuscript in preparation).

The main Vis color domain responsible for hyperpigmentation has been attributed to the shorter wavelengths of visible light (blue-violet light) [242,243]. The underlying mechanism leading to the activation of melanogenesis relies on a specific photoreceptor called opsin 3 and is able to sense blue light at the melanocyte membrane [243].

Similarly to the complementary effect of Vis with UVA1 in hyperpigmentation, Vis has also been shown to contribute to the pathophysiology of pigmentary disorders such as melasma [244] or actinic lentigines [245].

Overall, these findings, showing the involvement of Vis in skin alterations and its complementary or synergistic effect with UVA1, even at very small doses, could have implications for the treatment of photodermatoses and for photoprotection interventions [246].

## 5. Rationale for a Broadened Photoprotection, including UVA1

### 5.1. General Consideration of UV Photoprotection

The knowledge of the impact of both UVB and UVA rays in acute and chronic sun exposure argues for broad-spectrum UV-filtering photoprotection. Both the sun protection factor (SPF) related to erythemal prevention, mostly due to the shortest UVB wavelengths, and the UVA protection factor (UVA-PF) should thus be considered when choosing sunscreens to ensure that a broad spectrum photoprotection is granted in all solar exposure conditions and, even more specifically, in non-extreme daily exposures due to the prevalence of UVA [207,247]. Indeed, the effectiveness of the regular use of broad-spectrum sunscreen was demonstrated in several clinical studies, including those on the Nambour cohort, by reducing the development of precancerous actinic keratoses [248] and cancerous lesions (SCC and melanoma) [249] as well as clinical signs of photoaging [250] in the long term (>4.5 years). A review of two randomized clinical trials also supports the beneficial effects of the daily use of sunscreens on photoaging [251]. In addition to preventing sun damage, the daily use of a facial broad-spectrum photostable sunscreen may even visibly reverse the signs of existing photodamage, as shown by the follow-up of 32 subjects who applied daily facial SPF30 formulation for one year [252].

Broad spectrum filtration, with a ratio UVA PF/SPF of at least 1/3 and a critical wavelength > 370nm is therefore strongly recommended [139,207,253]. To determine the prediction of protection factors for long-term effects such as photoaging, a study aimed at analyzing the prevention of related early events at the molecular level (gene expression modulations) by a broad spectrum sunscreen with high SPF and UVA-PF [254]. In vivo, in human volunteers, the application of such sunscreen prevented the UVA radiation-induced transcriptional expression of genes involved in oxidative stress responses and dermal photoaging (heme oxygenase-1, superoxide dismutase-2, glutathione peroxidase, catalase, MMP-1) [254]. Many comparative studies demonstrated that, for a determined SPF value, a high level of UVA protection is required to obtain a better photoprotection. Such evidence was found regarding biological end points [255] as well as in clinical studies. Therefore, a better prevention of solar-induced hyperpigmentation could be obtained with a well-balanced sunscreen compared with a sunscreen with a low UVA protection level [256]. This was also observed for cumulative damages and P53 accumulation induced by repeated UV exposures [74,257]. Broad spectrum sunscreens were also shown to provide a higher efficacy for the prevention of PMLE [139,256] or the UV-induced suppression of DTH response to recall antigens [106,107,108,258,259].

### 5.2. Specific Attention on Long UVA Rays in Photoprotection

At present, state-of-the-art sunscreens can efficiently absorb part of UVA1, up to 370 nm, but not the longest UVA1 up to 400 nm. However, the specific deleterious effects of UVA1 in terms of DNA damage, photoaging, hyperpigmentation and immunosuppression, combined with the predominance of UVA1 rays in solar UV radiation and their presence in both acute and chronic conditions, strongly pleaded for broader photoprotection, taking into account these wavelengths, even for darker skins [207]. With regards to the common concern about vitamin D deficiency and the regular use of sunscreens, the enlargement of filtration within the UVA1 range should not have any impact due to the action spectrum of vitamin D synthesis, which peaks in the short UVB wavelengths range [260].

A first study addressing the question of the enlargement of photoprotection in the longest UVA1 wavelengths, up to 360 nm, showed that it was possible to inhibit the UVA1-induced expression of MMP1, responsible for the degradation of collagen fibers involved in the formation of wrinkles, as well as the expression of IL1 and IL6, in human dermal fibroblasts [261]. More recently, the enlargement of the profile of absorption by adding a UVA1 filtration up to around 400 nm led in vitro and in vivo to an improved skin photoprotection against an acute exposure to UVA1, as demonstrated in a pilot study with prototype formulations [262]. In reconstructed skin, a better protection compared to that of the control state-of-the-art sunscreen (with a photoprotection up to 370 nm and a similar SPF value) was indeed observed in terms of epidermal alterations and the depth of fibroblast disappearance as well as gene and protein expression from different functional families. In vivo, enlarging the UVA1 spectral absorption led to a significantly improved prevention of skin darkening, as revealed by colorimetric measurements and visual scoring comparisons versus a state-of-the-art sunscreen [262].

The positive efficacy results of this proof-of-concept study enabled the engagement of the development of a new UVA1 filter, Methoxypropylamino Cyclohexenylidene Ethoxyethylcyanoacetate (MCE, Mexoryl 400™), taking into account the regulatory and industrial feasibility constraints [263]. This filter has a very high absorption in the longest UVA1, with a peak at 385 nm. It is today listed in Annex VI of the EU-authorized UV filters. In a very recent paper, it was shown that the addition of MCE in reference formulas enlarged the profile of absorption up to 400 nm, leading to the coverage of the full UV spectrum, and improved UVA1 photoprotection by reducing UVA1-induced dermal and epidermal alterations at the cellular, biochemical and molecular levels as well as decreasing UVA1-induced pigmentation [264]. These data under controlled UVA1 exposures have been confirmed under a full UV (UVB and UVA) spectrum up to real sun exposure, supporting the benefit of full UVB and UVA profiles of absorption in natural sunlight exposures [265].

Other filters were also recently proposed to increase the profile of absorption within the UVA1 wavelengths. The TriAsorB™, a particulate filter, displays a very large, broad band of absorption, allowing for the enlargement of the sunscreen profile, but only moderately [266]. Another new organic filter, the bis-(diethylaminohydroxybenzoyl benzoyl) piperazine (BDBP), absorbs in the 350–425 nm range. In the SPF15 formulation, it has been shown to reduce the pigmentation induced by UV exposure at 385 nm [267].

### 5.3. Photoprotection Covering the Whole UV Spectrum and beyond UV: Vis or Pollutants

Exposure to other external aggressors, such as pollutants and Vis, was shown to induce independent, cumulative and synergistic effects with UVA1 rays (see Section 4). The combination of UVA1 and Vis, which can be received by sun exposure during approximately 2 h, can induce inflammation, immediate erythema [241] in lighter skin types and hyperpigmentation in darker individuals [268]. In light of those findings and of the influence of Vis on pigmentation and dyschromia, especially in darker skin phototypes, it is becoming increasingly evident that such broadened-spectrum sunscreens with a complete UV plus Vis protection is needed [207]. Today, to achieve an efficient VL protection, pigments such as iron oxides are added in the formulations. The development of methodologies to determine the level of protection in the VL range represents an emerging field to set up specific VL protection factors [239,269,270,271].

An additional benefit of photoprotection in the Vis range has been particularly demonstrated with regard to hyperpigmentary disorders, such as melasma with a regular use of a UV + Vis broad-spectrum sunscreen. [182,231,259,260,272]. Nonetheless, there is still a lack of awareness regarding photoprotection in darker-skinned populations, who are highly susceptible to pigmentary alterations [195,196].

Moreover, several mechanisms of pollution-derived hazards for the skin (oxidative stress, inflammation and metabolic impairments) might be amplified by the deleterious synergy of pollution and sun radiation, especially UVA [11,141,213]. An extensive photoprotection strategy within the full spectrum, including Vis, is thus required through adequate filtering systems and the addition of antioxidant ingredients, counteracting photopollution-induced oxidative stress [226,273].

## 6. Conclusions

In summary, a review of all the biological, cellular and clinical deleterious effects of UVA1, combined with the predominance of UVA1 rays in solar UV radiation, allows one to draw a solid rational for the need to protect against UVA1 radiation to prevent the short- and long-term effects of sun exposure on skin, particularly as daily photoprotection and for all phototypes. Moreover, considering the cumulative and synergistic effects of UVA1 with pollution or Vis, offering a filtration beyond 340 nm seems to be a requisite in order to confer efficient protection against solar damage associated with all the wavelengths, including UVA1 and Vis.

## Figures and Tables

**Figure 1 ijms-23-08243-f001:**
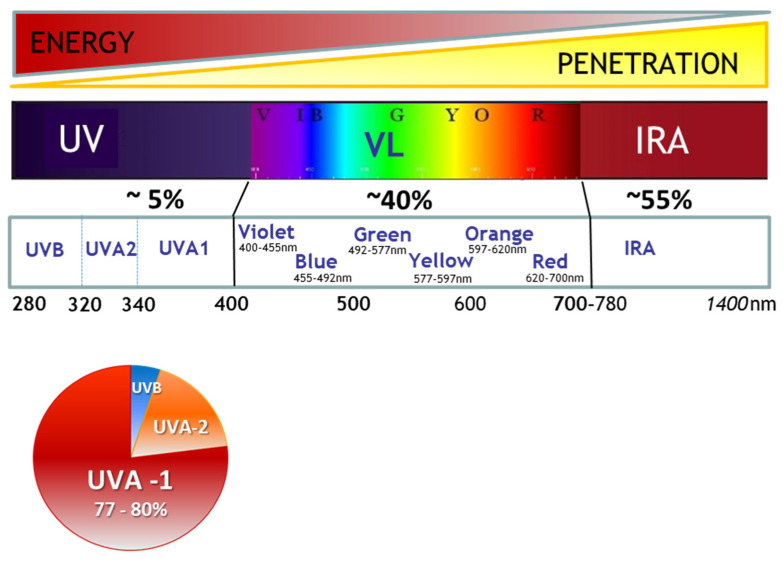
Solar spectrum. The three wavelength ranges along the solar radiation spectrum that are relevant for their impact on skin are ultraviolet (UV), visible light (Vis) and Infrared A (IRA). Energy and penetration properties are respectively lower and higher with increasing wavelengths. In the UV wavelengths domain UVA1 (long UVA), 340–400 nm can represent up to 80% of the total UV rays (for example, in the winter in temperate zones). UV: ultraviolet; IRA: Infrared A.

**Figure 2 ijms-23-08243-f002:**
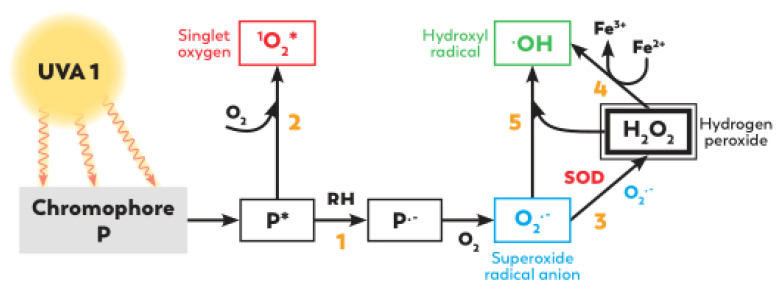
Simplified scheme of ROS formation after UVA exposure. Endogenous chromophore P can absorb UVA photons, leading to the formation of chromophores in photoexcited states P* such as the triplet state and initiating different reactions with molecular oxygen. In a type I reaction (1) (electron transfer), P* can react with an electron donor substrate (RH), leading to P^−^, which is able to react with oxygen, generating superoxide radical anion O_2_^−^ and a cascade of reactive reduction products. The spontaneous dismutation of O_2_^−^ leads to hydrogen peroxyde formation H_2_O_2_; this reaction is accelerated by superoxide dismutase SOD (3). H_2_O_2_ can then lead to hydroxyle radical OH^.^ formation via Fenton (4) and Haber–Weiss (5) reactions, thanks to electron donors such as Fe^2+^ or O_2_^−^, respectively. In a type II reaction (2) (energy transfer), the photoexcited sensitizer can react with molecular oxygen, leading to the formation of singlet oxygen (_1_O_2_*) capable of oxidizing a substrate. For more details, see [27].

**Figure 3 ijms-23-08243-f003:**
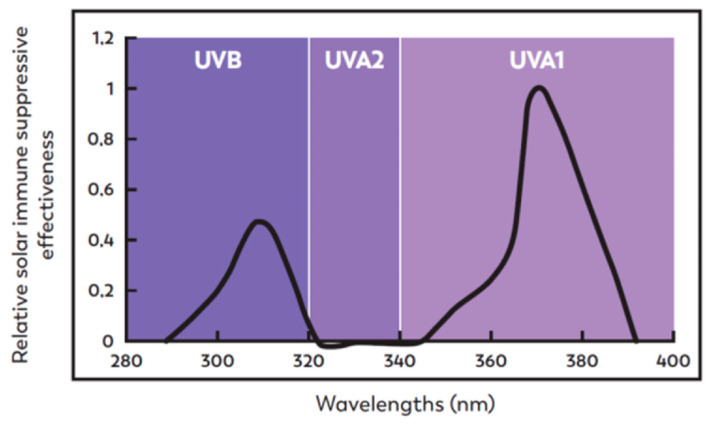
Solar effectiveness for immunosuppression in humans. Under natural daily moderate solar light exposure conditions, and taking into account the irradiance of the different UV domains, the graph shows that, apart from the well-known UVB contribution, UVA1 rays can have a predominant immunosuppressive effect. Adapted from [12].

**Figure 4 ijms-23-08243-f004:**
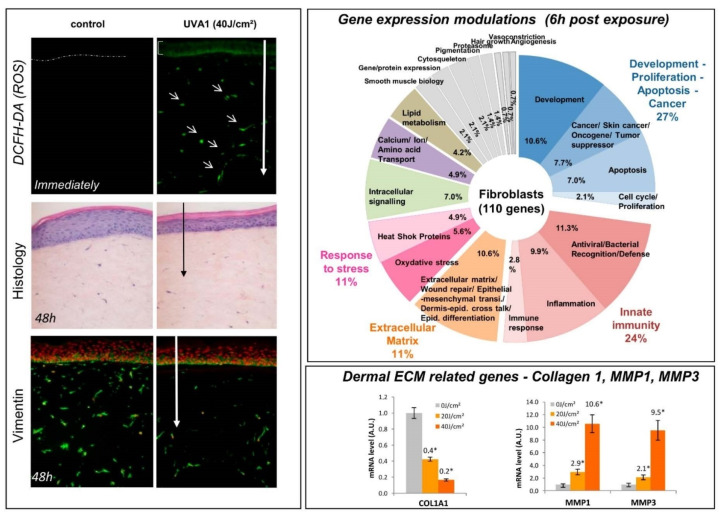
Dermal damages induced by UVA1 exposure in a reconstructed skin model. Due to their high penetration properties, UVA1 rays can reach the dermis. In a reconstructed skin model, it was possible to visualize the impact of UVA1 exposure thanks to the three-dimensional architecture of such model [33]. ROS generation can be observed within dermal fibroblasts immediately after the exposure to UVA1 (DCFH-DA staining), along with morphological alterations including the disappearance of dermal fibroblasts at 48 h (Histology and Vimentin staining). A full genome transcriptomic analysis revealed that 6 h post-UVA1 exposure a huge number of genes are modulated in fibroblasts. These modulated genes belong to various functional families such as Response to Stress, including oxidative stress, but also Extracellular Matrix, Innate Immunity and Development/Proliferation/Apoptosis/Cancer. To illustrate the contribution of UVA1-induced dermal alterations in the photoaging process, examples of modulated genes are illustrated with the downregulation of COL1A1 and the upregulation of MMP1 and MMP3. Small arrows: dermal fibroblasts containing ROS. Long arrows indicate the depth of UVA1 penetration. * Significant statistical difference (*p* < 0.05).

**Figure 5 ijms-23-08243-f005:**
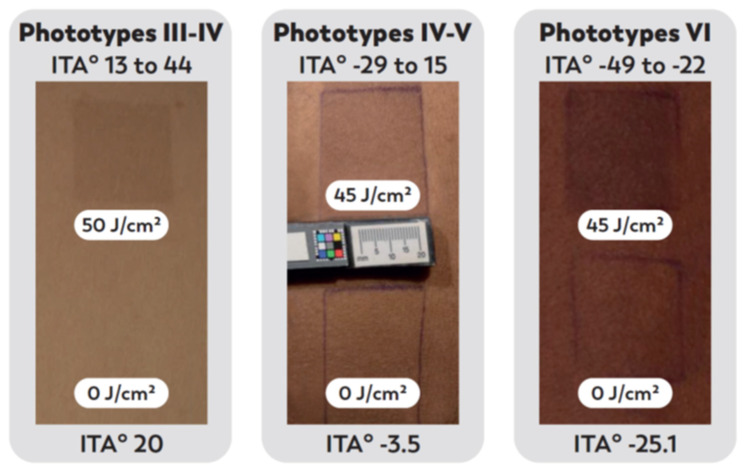
Hyperpigmentation induced by UVA1 exposure in vivo. Examples of UVA1-induced hyperpigmentation (PPD) 1.5–2 h after exposure to UVA1 in individuals with different skin color types, up to heavily pigmented skins. Inclusion criteria for Fitzpatrick phototypes and ITA° values (skin color type according to [193]) for the volunteers included in clinical studies are indicated at the top of one representative photograph. The specific ITA° values of the illustrated subjects are indicated at the bottom of the corresponding photograph.

**Figure 6 ijms-23-08243-f006:**
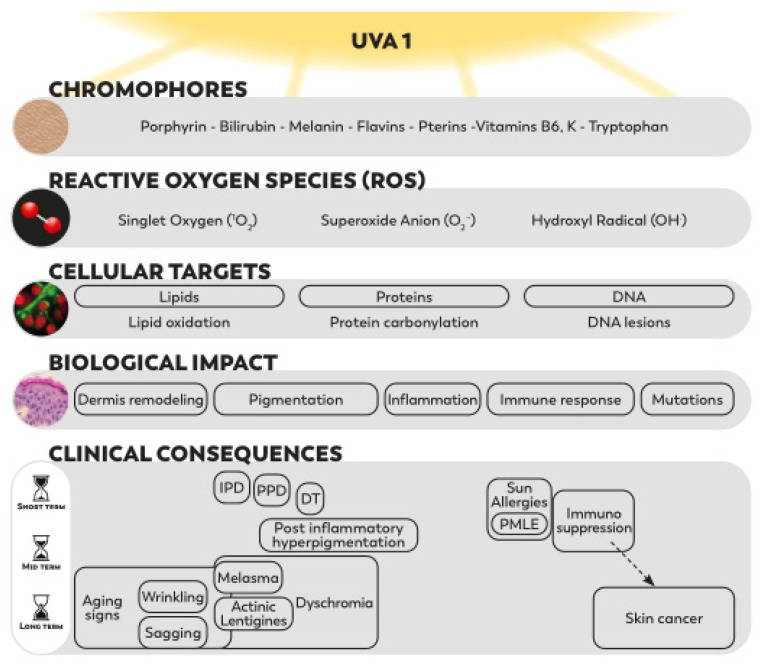
Schematic representation of UVA1 impacts on skin. The described effects of UVA1 exposure on skin are schematically represented, starting from the initial events with the absorption of UVA1 by chromophores and the generation of ROS to the biological and clinical consequences ranging from short-term to long-term effects. The dotted line indicates that photoimmunosuppression can contribute to skin cancer development.

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
