# Peer review of "The Damaging Effects of Long UVA (UVA1) Rays: A Major Challenge to Preserve Skin Health and Integrity"

_ijms, 2022, doi:10.3390/ijms23158243_

Round 1
Reviewer 1 Report
The work is in itself very detailed and complete, but I think that the objectives of the journal are not fulfilled here and therefore the work is not interesting for publication. Also considering its high impact factor, more innovative works should be published here. The review is rather a good summary, but clearly too long.
Author Response
We thank the Reviewer for the comments on the manuscript, especially the fact that he found it detailed and complete. We are sorry not to achieve to convince of the innovative aspect of this Review. The UVA1 wavelengths domain has been seen as negligeable for a long time and it is only recently that its impact in essential biological functions (DNA damages, immunity…) and the related clinical consequences (photocarcinogenesis, photoimmunosuppression) have been clearly demonstrated. Since the whole UV spectrum is considered as a full carcinogen by the WHO, and taking into account the high and constant exposure to such wavelengths, the photoprotection strategies should cover UVA1 rays up to 400nm. We would like to stress that it is only in 2022 that UV filters have been listed in the EU authorized list of UV filters for sunscreen use.
Following the recommendation of the Editor, we have submitted a revised version addressing the points raised by the other Reviewers.
Reviewer 2 Report
The authors have well-written review regarding the damaging effects of long UVA (UVA1) rays which is a major challenge to preserve skin health and integrity. The article is a comprehensive and well designed study. I hope the article will be interesting to scientific community and adds sufficient value to the literature regarding skin damage by UVA. I may pasting here the minor comments that the authors can follow to make it easier for the readers to follow it.
- English language should be improved.
- Authors should describe the methodology for this review i.e. inclusion criteria etc.
- In section 2.1.3. Cellular defense, reference number 29 is repeated consecutively in the same paragraph. It should be inserted at the end of the paragraph.
- In section 2.2.2. Protein carbonylation line number 231 and 234, reference number 38 is repeated consecutively in the same paragraph. It should be inserted at the end of the paragraph.
- In section 6. Conclusions, there shouldn’t be any reference in this section.

Author Response
We would like to thank the Reviewer for the comments and advice. We are pleased that the Reviewer found the manuscript comprehensive and well designed, as well as informative for readers.
1 The manuscript has been read by a native English speaker. Modifications have been made accordingly.
2: Methodology: This review was based on articles retrieved from Pubmed and Scopus bibliographic databases using “UVA1”, “longwave UVA”, “long UVA”, “skin” as main key words. This review does not pretend to be exhaustive, but we tried to gather the major facts related to clinical and biological cutaneous impact of UVA1 exposure.
3: Section 2.1.3. The reference 29 (now 30) has been added only at the end of the paragraph
4: Section 2.2.2. The reference 38 (now 39) has been added only at the end of the paragraph
5: Section 6: The reference has been removed from the conclusion section.
Reviewer 3 Report
The manuscript by Bernerd et al. an extensive and comprehensive review of all the biological, cellular, and clinical deleterious effects of UVA1, as well as their combined effects with air pollution and visible light.
It is a very well written manuscript, that shows an in-depth search of all the literature about the effects of UVA1, covering all the biological targets affected by this radiation, which are clearly presented to the reader and really show the need for broader protection range for sunscreens.
I only have to point out some small details:
Figure 1: the captions explains the meaning of HEV but that abbreviation is not found on the figure
lines 72-76: a reference is needed to support this claim
line 854-59: the complete reference must be on the reference list, not as citation on the text.
Throughout the manuscript, there are some citations (example: line 184 or 436) where the authors cite an author (example: Redmond et al.) but do not indicate the number to which that citation correspond on the reference list. Please correct
Also, it would be easier for the reader if there was a abbreviation list.
Author Response
We would like to thank the Reviewer for the comments and advice. We are pleased that the Reviewer found the Manuscript well written and comprehensive.
- We are sorry for the mistake and HEV has been removed from the legend of Figure 1.
- Lines 72-76: A reference to support the claim was added (ref 15).
- Line 854-59 (896 revised version): the citation has now been added as a reference (ref 253).
- Line 184 (188 revised version): Redmond et al was related to reference [34] (now ref 35), cited at the end of the paragraph.
- line 436 (456 revised version): Dumay et al. Is related to the reference [102] (now ref 111), cited at the end of the sentence.
Other reviewers asked to avoid repetition of the same reference within a paragraph, but to cite the reference only at the end of the paragraph.
We checked throughout the text that when an author was cited, the related reference was given at the end of the sentence or at the end of the paragraph.
- An abbreviation list has been added.
Reviewer 4 Report
The manuscript by Bernerd and coworkers is a comprehensive review describing the effects of long UVA on the different skin components. There are a number of deleterious effects which can be associated with clinical consequences and which are described in the review,
Authors should consider the following points aiming to further improve the manuscript:
1) Legend to figure 1: it is unnecessary long. It is not necessary to repeat what is already present in the text.
2) ine 295: Considering the effects of UVA on apoptosis of keratinocytes, a comparison of reconstituted skin vs normal skin can be done.
3) Line 451: The effects of UVA on lymphocytes should be better explored. For instance, due to the importance of T lymphocytes subsets in pathologic conditions, which is the biological effect of UVA on Th1 and Th2, Th17, Treg…..
4) Line 456: Urocanic acid should be mentioned in table or in paragraph 2.1
5) Line 496: the title of the subparagraph is not appropriate, since only solar urticaria is IgE-mediated. Therefore, the word “allergies” should be avoided and replaced with a more general term as “photodermatoses”.
6) Chapter 2.5.1: The role of elastases in the degradation of the elastic component should include also elastases released by neutrophils that are activated by UV exposure. Moreover, the paragraph should be implemented with the effects on GAGS, hyaluronate and CD44.
7) Line 698: It not clear why in the study performed on 1204 women, effects were independent on gender
8) The English language needs a revision to check for grammatical/typing errors.
Author Response
We would like to thank the Reviewer for comments and advice. We are pleased to see that the Reviewer found the Manuscript comprehensive.
1: Legend of figure 1: following the Reviewer’s comments, the legend has been shortened.
2: Line 295 (Paragraph started line 292 revised version): we have modified the paragraph by adding sentences and references to compare the in vivo and 3D in vitro conditions.
3: Line 451 (Paragraph started line 476 revised version): a paragraph about the effects of UVA1 on T lymphocytes subsets was added.
4: Line 456 (paragraph added line 111 revised version): Urocanic acid is now mentioned in paragraph 2.1.
5 Line 496 (529 revised version): according to the Reviewer’s comment, we have changed the title of the paragraph.
6: chapter 2.5.1: additional sentences and references to include the role of elastases as well as the modifications of GAGs (HA), and HA receptors such as CD44 have been added to the paragraph.
7: line 698 (740 revised version): we fully agree with the Reviewer. We are sorry for this mistake and modified the text accordingly.
8: English revision: manuscript has been revised by a native English speaker.
Reviewer 5 Report
The review: “The damaging effects of long UVA (UVA1) rays: a major challenge to preserve skin health and integrity” is an extensive collection of the knowledge from the different publications about deleterious effects of solar radiation, predominantly UVA, on the skin.
The authors described many aspects of the UVA induced skin damage. It would be advisable to add more on the following:
- The role of p53 in UVA induced skin damage
- The main differences and similarities in skin aging between UVA, Vis and UVB presented in the form of table
- UVA -induced Genetic mutations in skin and skin cells
- Clinical implications of vitamin D in sunscreen
Author Response
We would like to thank the Reviewer for comments and advice.
1: Sentences and references have been added with regards to p53 accumulation after UVA1 exposure. Apart from the link to DNA damage, the role of p53 in in melanogenesis has been added (lines 328 and 699).
2: We agree that a comparison between UVB and UVA mechanisms involved in skin ageing would be of interest. However, it seems quite difficult to add a short paragraph in this Review which is dedicated to the effects of UVA1. Although some differences, especially due to the different penetration properties, or mode of action with distinct chromophores for UVB and UVA, have been clearly shown, a rigorous comparative analysis would request a separate Review. Focusing on UVA1, it was important for us to highlight the fact that these wavelengths have a significant role, especially after chronic exposures. A sentence and references can be found at the beginning of the paragraph showing that chronic exposures to UVB or UVA similarly induce photoaging signs in mice (wrinkles, sagging). Concerning Visible light, except a very limited number of publications (in vitro data) suggesting a role in photoaging process (through ROS generation or MMP1 induction), there is not as yet any clinical proof of the implication of VL in photoaging.
3: A paragraph describing UVA-induced genetic mutations was added (line 348)
4: We agree with the Reviewer that the Vitamin D status/deficiency is a recurrent concern with regular use of sunscreens. However, the action spectrum for vitamin D synthesis is restricted to the short UVB range, and an efficient UVA1 photoprotection should therefore not have any impact. We have added a sentence related to that point in the Photoprotection section (line 921).